# Systematic Review with Meta-Analysis: Diagnostic Accuracy of Pro-C3 for Hepatic Fibrosis in Patients with Non-Alcoholic Fatty Liver Disease

**DOI:** 10.3390/biomedicines9121920

**Published:** 2021-12-15

**Authors:** Anne Linde Mak, Jenny Lee, Anne-Marieke van Dijk, Yasaman Vali, Guruprasad P. Aithal, Jörn M. Schattenberg, Quentin M. Anstee, M. Julia Brosnan, Mohammad Hadi Zafarmand, Dewkoemar Ramsoekh, Stephen A. Harrison, Max Nieuwdorp, Patrick M. Bossuyt, Adriaan G. Holleboom

**Affiliations:** 1Department of Vascular Medicine, Amsterdam University Medical Center, 1105 AZ Amsterdam, The Netherlands; a.vandijk2@amsterdamumc.nl (A.-M.v.D.); m.nieuwdorp@amsterdamumc.nl (M.N.); a.g.holleboom@amsterdamumc.nl (A.G.H.); 2Department of Epidemiology and Data Science, Amsterdam University Medical Center, 1105 AZ Amsterdam, The Netherlands; j.a.lee@amsterdamumc.nl (J.L.); y.vali@amsterdamumc.nl (Y.V.); m.h.zafarmand@amsterdamumc.nl (M.H.Z.); p.m.bossuyt@amsterdamumc.nl (P.M.B.); 3NIHR Nottingham Biomedical Research Centre, Nottingham University Hospitals NHS Trust and University of Nottingham, Nottingham NG7 2UH, UK; Guru.Aithal@nottingham.ac.uk; 4Metabolic Liver Research Program, Department of Medicine, University Medical Centre, Johannes Gutenberg University, 55131 Mainz, Germany; Joern.Schattenberg@unimedizin-mainz.de; 5The Newcastle Liver Research Group, Translational & Clinical Research Institute, Faculty of Medical Sciences, Newcastle University, Newcastle upon Tyne NE1 7RU, UK; quentin.anstee@newcastle.ac.uk; 6Internal Medicine Research Unit, Pfizer Inc., Cambridge, MA 02139, USA; julia.brosnan@pfizer.com; 7Department of Gastroenterology and Hepatology, VU University Medical Center, 1081 HV Amsterdam, The Netherlands; d.ramsoekh@amsterdamumc.nl; 8Radcliffe Department of Medicine, University of Oxford, Oxford OX3 9DU, UK; stephenharrison87@gmail.com

**Keywords:** fatty liver, biomarker, Pro-C3, collagen type III, fibrosis, liver disease

## Abstract

The prevalence and severity of non-alcoholic fatty liver disease (NAFLD) is increasing, yet adequately validated tests for care paths are limited and non-invasive markers of disease progression are urgently needed. The aim of this work was to summarize the performance of Pro-C3, a biomarker of active fibrogenesis, in detecting significant fibrosis (F ≥ 2), advanced fibrosis (F ≥ 3), cirrhosis (F4) and non-alcoholic steatohepatitis (NASH) in patients with NAFLD. A sensitive search of five databases was performed in July 2021. Studies reporting Pro-C3 measurements and liver histology in adults with NAFLD without co-existing liver diseases were eligible. Meta-analysis was conducted by applying a bivariate random effects model to produce summary estimates of Pro-C3 accuracy. From 35 evaluated reports, eight studies met our inclusion criteria; 1568 patients were included in our meta-analysis of significant fibrosis and 2058 in that of advanced fibrosis. The area under the summary curve was 0.81 (95% CI 0.77–0.84) in detecting significant fibrosis and 0.79 (95% CI 0.73–0.82) for advanced fibrosis. Our results support Pro-C3 as an important candidate biomarker for non-invasive assessment of liver fibrosis in NAFLD. Further direct comparisons with currently recommended non-invasive tests will demonstrate whether Pro-C3 panels can outperform these tests, and improve care paths for patients with NAFLD.

## 1. Introduction

Non-alcoholic fatty liver disease (NAFLD) is a spectrum of metabolic liver disease that is estimated to affect around 25% of the worldwide population [1]. Its incidence has swiftly risen in recent years, along with the increase in obesity and type 2 diabetes mellitus (T2DM) [1]. Although most individuals with NAFLD are in the first stage of simple steatosis without clinical symptoms, a considerable portion will progress into non-alcoholic steatohepatitis (NASH) and/or hepatic fibrosis, which may lead to cirrhosis and liver-related mortality [2,3]. Liver fibrosis is the main predictor of mortality in patients with NAFLD [4,5].

A key question in the field is to identify patients with advanced liver fibrosis in order to ensure preventative measures before irreversible liver damage occurs. Moreover, patients in advanced and potentially active and progressive stages of NAFLD are the ones most likely to benefit from anti-NASH drugs that are currently being developed [6].

Liver biopsy is the preferred method to assess disease activity and fibrosis stage [5,7,8]. However, it is invasive, time-consuming, costly and carries a small yet significant risk of complications [9]. These factors limit its use in patients with NAFLD [3]. Therefore, liver biopsy is currently only recommended for patients at an increased risk of NASH or advanced fibrosis, to provide prognostic information, to rule out other liver diseases or for enrollment in therapeutic clinical trials [10]. These limitations of liver biopsy hamper larger-scale identification and staging of patients with NAFLD, as does the general paucity of care paths and guidelines [11,12,13]. Consequently, a significant percentage of patients with NAFLD remain undiagnosed and not staged. Imaging by multiparametric MRI has been shown to be a good non-invasive tool to monitor NAFLD and predict liver-related outcomes [14,15,16], but multiparametric MRI is expensive and not readily available on a large scale to identify all patients with advanced stages of NAFLD in need of further care and surveillance. Transient elastography by FibroScan is another non-invasive imaging tool that can be used to estimate the degree of liver fibrosis, by using shear wave velocity combined with ultrasound to determine liver stiffness [17]. However, the FibroScan tool is not available at every center, especially not in primary care. Non-invasive biomarkers of NASH and fibrosis likely offer a safer, less expensive, and more broadly accessible and applicable alternative to liver biopsy.

In recent years, several biomarkers have been developed for NAFLD, including clinical risk calculation scores, individual blood-based markers, complex panels, and new imaging modalities [18]. Some, such as the Fibrosis-4 (FIB-4) score and the Enhanced Liver Fibrosis (ELF) test, are already implemented in some clinical guidelines, for excluding patients with severe disease in the first line of care, because of their high negative predictive values [10,19]. However, the specificity of FIB-4 becomes low above the age of 65 [20,21] and the commercially available ELF test is quite costly, limiting its implementation.

A relatively new biomarker to be considered and tested for NAFLD is Pro-C3, an enzyme-linked immunosorbent assay (ELISA) towards the N-terminal propeptide of type III collagen. This ELISA method measures ADAMTS2-mediated cleavage of the propeptide from type III collagen during fibrillar assembly, making the test outcome indicative of active fibrogenesis [22]. Since a dense fibrillar collagen band can block the passage of nutrients and oxygen through the interstitial extracellular matrix, this is an important structure in the development of fibrosis [23]. Pro-C3 was developed to assess the formation of type III collagen in different pathologies, hence not specifically for NAFLD [24]. This marker has been studied in patients with fibrosis related to chronic hepatitis C [25], after liver transplantation, tumor fibrosis in pancreatic ductal adenocarcinoma [26] and lung fibrosis in patients with systemic sclerosis [27]. Several studies have assessed the performance of Pro-C3 in NAFLD, mostly as a single marker and sometimes as part of a diagnostic biomarker panel [28,29,30,31]. To date, it is unknown for which target condition within the NAFLD spectrum; in which context of use (i.e., in primary, secondary or tertiary care); and at what diagnostic threshold, that Pro-C3 has optimal performance.

Thus far, the number of studies that evaluated the diagnostic accuracy of Pro-C3 is limited, and different levels of accuracy are reported. The largest study to investigate the diagnostic accuracy of Pro-C3 included 449 subjects, in a setting with a relatively high prevalence of advanced fibrosis (37%) [28]. A meta-analysis of the accuracy of Pro-C3 based on all available studies can provide a more comprehensive evidence-base. We performed a systematic review and meta-analysis to summarize the clinical performance of Pro-C3 in detecting significant fibrosis (F ≥ 2), advanced fibrosis (F ≥ 3), cirrhosis (F4), and NASH in patients with NAFLD.

## 2. Materials and Methods

This work has been conducted as part of the international multi-center LITMUS project (Liver Investigation: Testing Marker Utility in Steatohepatitis), aiming to develop and validate a defined set of biomarkers for detecting of NASH and NAFLD-related fibrosis. The protocol of the full systematic review can be accessed on PROSPERO: CRD42018106821. The current report was prepared using the PRISMA-DTA statement (Appendix A) [32].

### 2.1. Literature Search

We developed a sensitive search strategy to identify reports on the diagnostic accuracy of Pro-C3 in patients with NAFLD. The search strategy encompassed words in the title or abstract as well as in the full record and used the Medical Subject Heading (MeSH) terms. In January 2020, the following databases were searched: MEDLINE (via OVID), EMBASE (via OVID), PubMed, Science Citation Index and CENTRAL (the Cochrane Library). The full search strategy can be found in Appendix A. To identify additional studies, we hand searched the list of references of eligible study reports and contacted our LITMUS partners for any studies that might have been missed by our search strategy. The search was updated in July 2021. All available records that fulfilled the search criteria to that date were screened.

### 2.2. Eligibility Criteria

Studies that included patients with NAFLD and undertook both Pro-C3 testing and a liver biopsy as the reference standard, published in peer-reviewed journals or identified conference abstracts/presentations, were eligible. Prognostic studies were excluded. We made no further restrictions on language or study design. Only studies of adult patients (≥18 years) with NAFLD were included. Study groups consisting of patients with mixed etiologies were excluded if data for patients with NAFLD were not reported separately. Studies that included patients with co-existing liver disease (e.g., both NAFLD and hepatitis B) were excluded. Potential overlap of patient groups between studies was thoroughly investigated and checked with the authors. In case of overlapping patient groups between studies, the largest study was selected.

Only studies using a liver biopsy as the clinical reference standard were eligible. The target conditions were NASH, significant liver fibrosis (F ≥ 2), advanced liver fibrosis (F ≥ 3), and cirrhosis (F4). The index test was the Pro-C3 ELISA, developed by Nordic Biosciences in 2013 [24]. Time between Pro-C3 measurements and liver biopsies could not be more than 6 months for a study to be eligible.

Studies that provided data on true positive (TP), false positive (FP), true negative (TN) and false negative (FN) Pro-C3 test results, or data that allowed us to reconstruct the classification table, were eligible for inclusion in the meta-analysis. In case studies did not report a threshold value for Pro-C3, or did not report sufficient information to reconstruct classification tables to calculate diagnostic accuracy estimates, authors were contacted to provide the required information. All studies from authors who decided not to or failed to provide these data within four months were excluded from the meta-analysis.

### 2.3. Study Selection and Data Extraction

Two authors (ALM and AvD) independently screened the identified titles and abstracts for potentially eligible studies. The same two authors subsequently evaluated full-text articles reporting potentially eligible studies to make final decisions about inclusion. The title and abstract screening phases were conducted using Rayyan QCRI [33]. Any conflict of judgment was discussed and resolved between the two primary reviewers; when inconclusive, a third reviewer’s judgment (JL) was decisive.

The following data were extracted from the included studies by the first reviewer (ALM) and cross-checked by the second reviewer (AvD): study characteristics, study group characteristics, reference test features, index test features, and numbers needed for reconstructing 2 × 2 classification tables (TP, FP, TN, FN).

### 2.4. Risk of Bias and Applicability Assessment

The risk of bias and applicability to our review question of each included study were independently assessed by two reviewers (ALM and AvD) using the QUADAS-2 (Quality Assessment of Diagnostic Accuracy Studies) tool in Review Manager 5.4 [34,35]. Any conflict of opinion was discussed and resolved between them, or discussed with a third reviewer (JL or YV). For each of the domains of the QUADAS-2 tool, the risk of bias for each study was assigned a judgment of ‘low risk’, ‘unclear risk’ or ‘high risk’. Concerns about applicability to the review question were classified in the same manner.

### 2.5. Statistical Analysis

Forest plots summarizing sensitivity, specificity and threshold values reported in each study were designed using Review Manager 5.4 [35].

We conducted a meta-analysis of the accuracy of Pro-C3 in detecting the respective target conditions (i.e., significant fibrosis, advanced fibrosis, NASH or cirrhosis), whenever three or more studies reporting accuracy data for this target condition were available. A bivariate logitnormal random effects model was used to compute summary measures of the diagnostic accuracy of Pro-C3, in terms of sensitivity and specificity and area under the receiver operating characteristic curve (AUC) [36]. From studies that reported accuracy data at more than one threshold, one threshold was selected for the analysis: either the point maximizing the Youden Index or, if no Youden Index was calculated, the threshold closest to those of the other included studies.

Summary receiver operating characteristic (SROC) curves were constructed using the estimates from the bivariate random effects model, to illustrate the overall diagnostic accuracy of Pro-C3 for each target condition, including 95% confidence intervals. We further calculated 95% prediction intervals to assess between and within study heterogeneity. All analyses were performed in R version 3.6.3, using the ‘mada’ package [37,38]. Additional bootstrapping was done using the ‘dmetatools’ package to calculate the 95% confidence interval around the AUC [39].

Publication bias was not assessed in this systematic review, as no reliable methods are available for evaluating publication bias in diagnostic accuracy studies [40].

### 2.6. Sensitivity Analysis

Sensitivity analyses were conducted by repeating the meta-analysis excluding one primary study that included only patients with T2DM [41] and one abstract-only study [42].

## 3. Results

### 3.1. Study Characteristics

From the 35 records on Pro-C3 which were identified in the search, eight studies were eligible for inclusion in the systematic review. The study flow diagram is shown in Figure 1. Seven of the included studies were full text reports [28,29,30,41,43,44,45] and one was an abstract [42].

The study characteristics are presented in Table 1. One study had two separate groups for the current analysis because the characteristics and outcomes of the discovery and validation cohorts were reported separately [43]. Seven out of the eight studies were designed as diagnostic accuracy studies, while one was an exercise intervention trial that reported Pro-C3 values and liver biopsies at baseline [44]. Included patients were mostly from Western countries (United States and Europe); two studies also included participants from Asia and Australia [29,30].

The mean age of individuals was 54 years (SD 14.2) and the average percentage of male participants was 55%. Of note, seven out of eight studies had recruited patients from secondary or tertiary care only. One study included solely patients with T2DM [41], whereas the percentage of patients with T2DM in the other studies ranged from 27% to 48%.

Seven studies reported the accuracy of Pro-C3 in detecting significant fibrosis, eight for advanced fibrosis, three for (fibrotic) NASH, and two for cirrhosis or cirrhotic NASH. All studies used the NASH CRN criteria for histological scoring of liver biopsies, the clinical reference standard. Appendix A details what additional information was received from authors of primary studies.

### 3.2. Risk of Bias Assessment

The overall judgment of methodological quality of each of the included studies is shown in Appendix A. Overall, the risk of bias in the included studies was low. One study was deemed at a high risk of bias for patient selection for the purpose of this systematic review because of inappropriate exclusions [44]. Another abstract-only study did not describe patient selection in sufficient detail [42]. Two studies were deemed at high risk of bias for the index test, because of unclear reporting of the interpreters’ blinding, of how the Pro-C3 threshold was determined, or because the threshold was not pre-specified [29,41]. There were few applicability concerns; the only high concern judgment here was based on the inclusion solely of patients with T2DM in one primary study [41].

### 3.3. Accuracy of Pro-C3 in Detecting Significant Fibrosis

The meta-analysis of the accuracy of Pro-C3 in detecting significant liver fibrosis (F ≥ 2) comprised data from seven primary studies on 1568 patients with NAFLD in total. The proportion of study participants with significant liver fibrosis in the included studies ranged from 36% to 74%.

Appendix A shows the forest plot of the diagnostic accuracy data from primary studies in detecting significant fibrosis. As shown, included studies used different positivity threshold values for Pro-C3, ranging from 9.7 to 20.9 ng/mL.

The SROC curve in Figure 2 depicts the sensitivity versus the false positive rate (1–specificity) of Pro-C3 for detecting significant fibrosis. As shown, the estimated mean specificity and sensitivity of Pro-C3 were 79% (95% CI 0.71–0.86) and 68% (95% CI 0.50–0.82), respectively. The AUC was 0.81 (95% CI 0.77–0.84) for detecting significant fibrosis.

### 3.4. Accuracy of Pro-C3 in Detecting Advanced Fibrosis

Nine groups from eight primary studies were available for meta-analysis of the accuracy of Pro-C3 in the detection of advanced fibrosis. In total, these included 2058 patients with NAFLD. The proportion of study participants with advanced fibrosis in the included studies ranged from 17% to 47%.

The forest plot in Appendix A provides an overview of the diagnostic accuracy data of Pro-C3 in detecting advanced fibrosis. The threshold values ranged from 12.7 to 21.3 ng/mL: six studies employed a threshold around 15 ng/mL, while the three remaining studies used a threshold around 21 ng/mL. Figure 3 shows the SROC curve of Pro-C3 in the detection of advanced fibrosis. For this target condition, the specificity and sensitivity were similar: 73% (95% CI 0.65–0.80) and 72% (95% CI 0.62–0.81), respectively. The summary estimate of the AUC of Pro-C3 in detecting advanced fibrosis was 0.79 (95% CI 0.73–0.82).

One point in the SROC curve is depicted at a sensitivity of 1.0, since there were no false negatives among the 56 patients with advanced fibrosis included in the study from Knöchel and colleagues [42]. Because of the small number of participants in this study, this had no substantial influence on the summary curve.

### 3.5. Accuracy of Pro-C3 Detecting Non-Alcoholic Steatohepatitis (NASH) or Cirrhosis

One study reported accuracy data for NASH, and reported a sensitivity of 55% and sensitivity of 82% at a threshold of 14.7 ng/mL [30]. Two studies reported accuracy data of Pro-C3 in detecting fibrotic NASH, defined as NAS ≥ 4 plus F ≥ 2, and estimated sensitivity at 66% and 68%, and specificity at 68% and 73%, for thresholds of 14.5 and 12.6, respectively [28,45]. For cirrhosis or cirrhotic NASH, defined as NAS ≥ 4 plus F4, sensitivities of 90% and 76%, and specificities of 57% and 63% were reported, at thresholds of 15.6 and 16.5 ng/mL, respectively [28,29]. Since the classification of (fibrotic) NASH differed between the three studies and only two studies reported accuracy data for cirrhosis, insufficient data were available for meta-analyses of the diagnostic accuracy of Pro-C3 for these target conditions.

### 3.6. Sensitivity Analyses

We performed a sensitivity analysis excluding the one study that included only patients with T2DM [41]. We observed no major differences in sensitivity or specificity for the detection of either significant or advanced fibrosis with or without the T2DM study. For significant fibrosis, the sensitivity analysis yielded an AUC of 0.81; for advanced fibrosis, the AUC was 0.78.

Another sensitivity analysis was performed excluding the data from the included abstract [42] as non-peer-reviewed data may be less reliable. The AUCs in this sensitivity analysis remained the same as the main analysis (0.81 for significant fibrosis and 0.79 for advanced fibrosis) and there were no major differences in sensitivity or specificity.

## 4. Discussion

### 4.1. Main Results

In this systematic review, we summarized the evidence on the accuracy of Pro-C3 in detecting target conditions within the NAFLD spectrum: significant fibrosis (F ≥ 2), advanced fibrosis (F ≥ 3), or NASH. We found that Pro-C3 had an overall AUC around 0.80 in detecting significant and advanced fibrosis in patients with NAFLD. Few studies have reported on the accuracy of Pro-C3 in detecting NASH or cirrhosis.

Currently, clinical guidelines recommend using non-invasive biomarkers such as FIB-4 and the ELF test to predict NAFLD fibrosis [10,19,46]. A recent systematic review showed that the AUC of the ELF test in detecting advanced fibrosis is 0.83 [47]. For FIB-4, a study with a large sample size reported an AUC of 0.76 in detecting advanced fibrosis [48]. The results of the current systematic review and meta-analysis, therefore, indicate that the diagnostic accuracy of Pro-C3 is comparable to these tests that are already mentioned in clinical guidelines, and should be seen as an important candidate in improving non-invasive diagnostics for NAFLD fibrosis. We suggest that Pro-C3, or a panel incorporating Pro-C3, may be used as an adjunct test to the widely available FIB-4 test in a two-tiered screening approach. It could replace transient elastography, especially in clinics where FibroScan is not available, although the accuracy of this suggested approach should still be investigated.

### 4.2. Test Performance in Detecting Liver Fibrosis

Pro-C3 is a measurement of collagen cleavage during active fibrogenesis. Other collagen cleavage particles have been evaluated in diagnosing hepatic fibrosis as well, such as PIIINP, a component of the ELF test panel. The exact epitope of PIIINP is not known, and PIIINP can be a marker of both formation and degradation of extracellular matrix [23,24]. In contrast, the Pro-C3 ELISA antibody specifically binds to the ADAMTS2 cleavage site of the N-terminal propeptide of type III collagen. Pro-C3 can thus be expected to be more specific than PIIINP for fibrogenesis. Indeed, our analyses indicate that the specificity of Pro-C3 could be high: the summary estimate was 79% (95% CI 0.71–0.86) for detecting significant fibrosis. There is some uncertainty to this result, reflected in the 95% confidence area around the mean and the 95% prediction region. False positivity of Pro-C3 can be expected in patients with fibrogenesis in other tissues, since type 3 collagen is not liver-specific, but also can be generated in muscle tissue or when lung fibrosis occurs [24,27]. We found that the mean sensitivity of Pro-C3 in detecting significant and advanced fibrosis was 68% (95% CI 0.50–0.82) and 72% (95% CI 0.62–0.81), respectively. The relatively low sensitivity may be explained by the fact that Pro-C3 is a marker of active collagen turnover, while liver biopsy as the comparison standard only gives a snapshot of fibrosis, the result of a chronic process of damage and repair in NAFLD. Therefore, based on a liver biopsy, one is likely unable to tell whether patients have been at a certain fibrosis stage for a long time or have recently progressed there. It will be highly interesting and clinically relevant to investigate the predictive potential of Pro-C3 in identifying patients that will progress in fibrosis stage, especially since it has been shown that Pro-C3 has this ability in chronic hepatitis C [49]. Bril and colleagues already reported that changes in Pro-C3 could identify patients that improved by 2 or more points in fibrosis score after treatment with pioglitazone and/or vitamin E, with an AUC of 0.85 [41]. Recent results from the phase 2B CENTAUR study on cenicriviroc indicate that Pro-C3 as a biomarker may have prognostic value as well [50].

### 4.3. Test Performance in Detecting NASH

From the limited data available, we observed low performance of Pro-C3 in detecting NASH; sensitivity ranged from 55% to 68%, and specificity ranged from 68% to 82%. This may be attributed to the paradigm that fibrosis progression results from repetitive periods of inflammation, as in NASH, alternated with periods of ‘repair’ of the extracellular matrix (ECM) [51]. The three studies that evaluated Pro-C3 accuracy for NASH used different criteria to establish NASH and, therefore, cannot be accurately compared [28,30,45]. All three studies used a NAFLD activity score (NAS) of ≥4 with at least 1 point each for steatosis, hepatocyte ballooning and hepatic inflammation to define NASH. However, the studies by Boyle and Erhardtsen added a fibrosis stage of ≥F2 to these criteria, and thus were purely looking at active stages of NASH.

### 4.4. Panels Including Pro-C3

Several studies included in this systematic review also assessed the diagnostic accuracy of a combination of Pro-C3 measurements with clinical data such as age, body mass index (BMI), T2DM status, and routine clinical tests, such as platelet count. Daniels and colleagues established the ADAPT algorithm, finding an increased AUC for detecting advanced fibrosis with ADAPT compared to Pro-C3 alone of 4% to 0.87 [29]. Boyle and colleagues found their FIB-C3 algorithm could increase diagnostic accuracy for detecting advanced fibrosis from an AUC of 0.76 for Pro-C3 alone to 0.85 for FIB-C3, similar to the performance of the ADAPT score in their study group. A simplified version of this panel that would be more easily calculated in a clinical setting, ABC3D, performed similarly to FIB-C3 with an AUC of 0.83 [28]. An overview of the accuracy of these panels for NAFLD fibrosis is given in Appendix A. The incorporation of Pro-C3 into a diagnostic panel offers the attractive combination of a direct test of active collagen turnover with clinical variables widely known to increase the risk of advanced disease. All three diagnostic panels evaluated by Boyle and colleagues (i.e., FIB-C3, ABC3D and ADAPT) were shown to outperform FIB-4 in their study group [28]. For clinical use, we think these panels show potential over Pro-C3 alone and should be further validated.

### 4.5. Strengths and Limitations

This study presents the first meta-analysis of the diagnostic accuracy of Pro-C3 with liver biopsy as the reference standard. Results from more than 1500 patients were included in our meta-analysis of significant fibrosis and more than 2000 in that of advanced fibrosis.

An aspect that merits consideration is that we were unable to evaluate the performance of Pro-C3 in a primary care setting. In a setting with a low prevalence of advanced disease one can expect to find undiagnosed patients with potentially advanced NAFLD that could benefit most from early detection.

From the included articles in this review, we were not able to acquire enough accuracy data at different threshold values to report accuracy at different threshold values. Future research should evaluate the diagnostic accuracy of Pro-C3 across several pre-determined thresholds with respect to the context of use, not only at the threshold identified by Youden Index, which attributes equal importance to false negatives and false positives, a condition that does not reflect the respective clinical consequences. To evaluate the potential of a marker for use in primary, secondary or tertiary care centers, the sensitivity and specificity of the test should be evaluated in the respective setting and false positives and false negatives should be differentially weighted.

It should be pointed out that availability of the Pro-C3 ELISA is still limited. It is exclusively produced by Nordic Biosciences, and at present has solely been used for research purposes. Before this test can be recommended for clinical use, its availability should be broadened. Pro-C3 tests are expected to soon become available worldwide on the Roche COBAS platform, and an assessment costs around 40–50 euros. Of note, the recent study by Erhardtsen and colleagues investigated the robustness and analyte stability of the Pro-C3 assay and found it passed the predefined acceptance criteria for precision [45].

### 4.6. Conclusion and Recommendations for Future Research

This review shows that Pro-C3 holds potential for the non-invasive assessment of NAFLD fibrosis. We recommend focusing future studies into Pro-C3 on both diagnostic and prognostic evaluations, and to recruit patients in the intended use setting. Firstly, it is necessary to further evaluate the diagnostic potential of Pro-C3, focusing on the diagnostic panels that incorporate Pro-C3 measurement. Further direct comparisons between these Pro-C3 panels and currently recommended tests, such as FIB-4 and ELF, will resolve whether Pro-C3 panels can, indeed, outperform these tests. Secondly, it is appealing to assess the prognostic potential of Pro-C3 in paired liver biopsy studies for progression in fibrosis stage and treatment response. Both of these directions may provide a basis for future guidelines, and for improvement of care paths for patients with NAFLD.

## Figures and Tables

**Figure 1 biomedicines-09-01920-f001:**
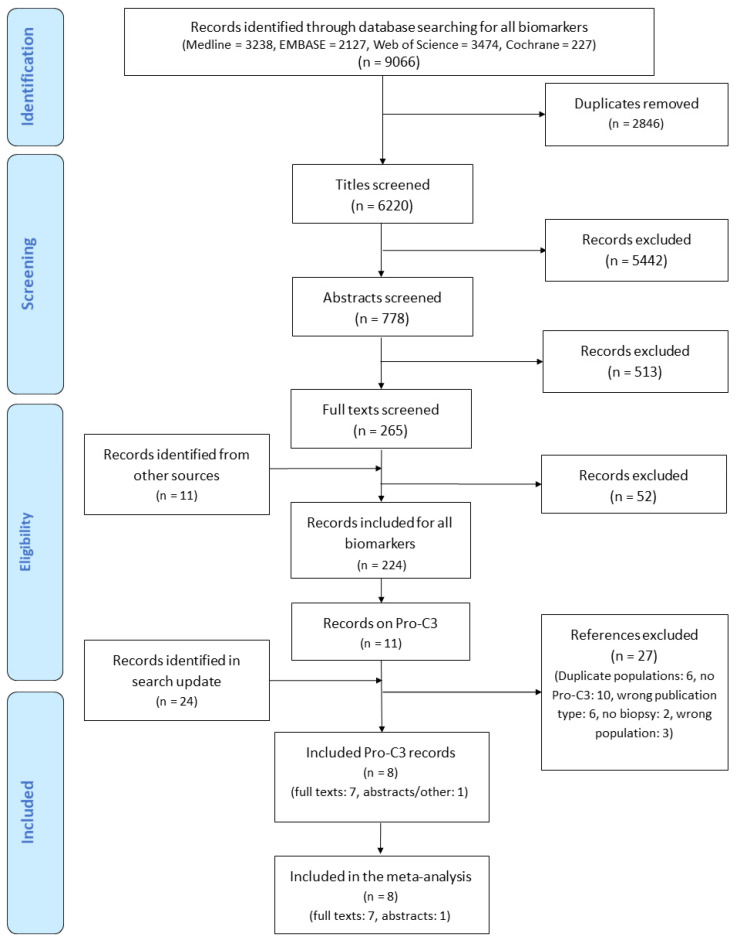
Flow diagram of studies included in the systematic review and meta-analysis.

**Figure 2 biomedicines-09-01920-f002:**
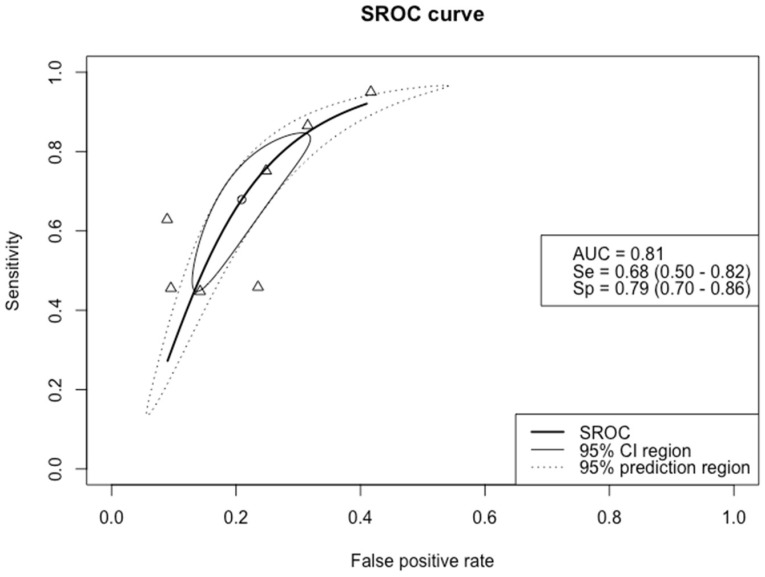
Summary receiver operating characteristic (SROC) curve of the diagnostic accuracy of Pro–C3 in detecting significant fibrosis. The solid ellipse depicts the 95% confidence interval region of diagnostic accuracy data of Pro–C3 in the included studies; the dotted ellipse depicts the prediction region in which 95% of future diagnostic accuracy study estimates of Pro–C3 will fall. Triangles represent diagnostic accuracy estimates from each included study; circle represents the Youden Index threshold value. AUC = area under the receiver operating curve, Se = sensitivity, Sp = specificity.

**Figure 3 biomedicines-09-01920-f003:**
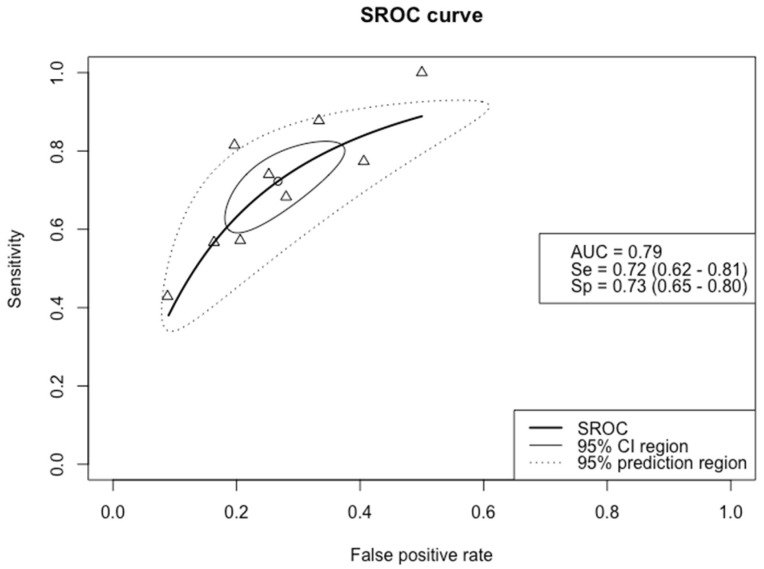
SROC curve of the diagnostic accuracy of Pro–C3 in detecting advanced fibrosis. The solid ellipse depicts the 95% confidence interval region of diagnostic accuracy data of Pro–C3 in the included studies; the dotted ellipse depicts the prediction region in which 95% of future diagnostic accuracy study estimates of Pro–C3 will fall. Triangles represent diagnostic accuracy estimates from each included study; circle represents the Youden Index threshold value. AUC = area under the receiver operating curve, Se = sensitivity, Sp = specificity.

**Table 1 biomedicines-09-01920-t001:** Characteristics of included studies.

Study ID	Country	Setting	Population	N (% Male)	Mean Age	BMI (SD)	Target Conditions	DM	AST (U/L)	ALT (U/L)
Daniels 2019 [29]	Australia, UK, Japan	Secondary and tertiary care	Biopsy-confirmed NAFLD	239 (56%)	52.2	33.6 (7.7)	F ≥ 2; F ≥ 3; F4	37%	49.6 (34.4) *	72.2 (54.6) *
Boyle 2019 [28]	7 European countries	Tertiary care	Suspected NAFLD	449 (59%)	52.0	32.6 (6.8)	F ≥ 3; NASH + F ≥ 2; NASH + F4	48%	47.0 (26.0)	69.0 (41.0)
Huber 2019 [44]	Germany	Secondary or tertiary care	Biopsy-confirmed NAFLD	27 (66%)	41.0 †	30.8 (5.1)	F ≥ 2; F ≥ 3	27%	NR	NR
Luo 2018 Discovery [43]	USA	Secondary or tertiary care	Suspected or biopsy-confirmed NAFLD	164 (32%)	53.3	NR	F ≥ 2; F ≥ 3	NR	46.8 (21.3) *	59.8 (38.1) *
Luo 2018 Validation [23]	USA	Secondary or tertiary care	Biopsy-confirmed NAFLD	41 (32%)	50.1	NR	F ≥ 2; F ≥ 3	37%	71.3 (50.6) *	98.3 (57.5) *
Nielsen, Leeming 2021 [30]	USA, Australia, Belgium, France, Germany, Hong Kong, Italy, Poland, Spain, UK	Secondary or tertiary care	Biopsy-confirmed NAFLD	517 (52%)	55.2 †	32.7 †	F ≥ 2; F ≥ 3; NASH	40%	34.8 †	47.1 †
Bril 2019 [41]	USA	Primary and tertiary care	Suspected NAFLD	125 (87%)	58.7	34.4 (4.6)	F ≥ 2; F ≥ 3	100%	40.4 (23.1)	53.6 (35.6)
Knöchel 2021 [42]	Sweden	Secondary or tertiary care	Biopsy-confirmed NAFLD	56 (71%)	61.0	29.1 (4.7)	F ≥ 2; F ≥ 3	NR	NR	NR
Erhardtsen 2021 [45]	UK and Germany	Secondary and tertiary care	Biopsy-confirmed NAFLD	215 (52%)	56.0	33 †	F ≥ 2; F ≥ 3; NASH + F ≥ 2	47%	48.5 †	64.0 †

* not documented for all patients; † = median, not mean; NAFLD = non-alcoholic fatty liver disease, NASH = non-alcoholic steatohepatitis, AST = aspartate aminotransferase, ALT = alanine aminotransferase, BMI = body mass index, NR = not reported, SD = standard deviation, DM = diabetes mellitus.

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
