# Peer review of "Systematic Review with Meta-Analysis: Diagnostic Accuracy of Pro-C3 for Hepatic Fibrosis in Patients with Non-Alcoholic Fatty Liver Disease"

_biomedicines, 2021, doi:10.3390/biomedicines9121920_

Round 1

Reviewer 1 Report

The manuscript describes a meta-analysis using data from published studies on the diagnosis accuracy of the biomarker Pro-C3 to detect significant fibrosis, advanced fibrosis or steatohepatitis (NASH) in patients with non-alcoholic fatty liver disease (NAFLD). Data from seven full text reports and one abstract were included. A total of 1,568 patients were included in the analysis to assess accuracy of Pro-C3 in detecting significant fibrosis, and 2,058 patients were included in the analysis to assess accuracy of Pro-C3 in detecting advanced fibrosis. The authors conclude that Pro-C3 has an overall area under the curve of approximately 0.8 to detect significant and advanced fibrosis in patients with NAFLD. No analysis could be performed on NASH, due to discrepancies on the classification of fibrotic NASH between studies. The authors indicate that Pro-C3 provides information on active collagen turnover occurring at the time of measurement, and could be used as a non-invasive assessment of NAFLD fibrosis, in addition to other established biomarkers. Overall, the conclusions are supported by the data presented, and this study provides valuable information in the field of hepatology and NAFLD.

Comments:

-Eligibility criteria. The basis for study inclusion is a bit vague. Lines 123-124 state that only studies of adult patients with clinically suspected or biopsy-proven NAFLD were included. Line 131 indicates only studies using liver biopsy as a clinical reference standard were included. The two statements are confusing, and need clarification.

Also, Table 1 includes patients with ‘suspected NAFLD’. However, there is no information on what clinical parameters were used to diagnose these patients with NAFLD. This information could be provided in the table.

-Lines 136-142 are unclear. What data used was taken only from published data, and which studies required additional data that was obtained by contacting the authors? This could be added as a separate table in the supplementary data section.

-Reference 44 is an abstract, and therefore, this is not peer-reviewed published information. The data could be completely removed from the meta-analysis, or if included, a statement could be added indicating that the removal of these data does not affect the results shown in the Figures as well as the conclusions of the meta-analysis.

Author Response

Thank you for your comments and suggestions. Below here, we respond to the mentioned suggestions point by point.

Point 1: Eligibility criteria. The basis for study inclusion is a bit vague. Lines 123-124 state that only studies of adult patients with clinically suspected or biopsy-proven NAFLD were included. Line 131 indicates only studies using liver biopsy as a clinical reference standard were included. The two statements are confusing, and need clarification.

Also, Table 1 includes patients with ‘suspected NAFLD’. However, there is no information on what clinical parameters were used to diagnose these patients with NAFLD. This information could be provided in the table.

Response 1: Thank you for pointing out this confusion, we agree that this should described more clearly. The studies included in our meta-analysis were allowed to include patients into their study cohort that were either already confirmed to have NAFLD based on a liver biopsy, or were suspected to have NAFLD based on non-invasive methods such as an abdominal ultrasound or serum liver enzymes. Still, all patients needed to undergo a liver biopsy during the study within 6 months of the blood withdrawal for Pro-C3 testing.

We have removed the words “clinically suspected or biopsy-proven” from line 128 to avoid this confusion.

In Table 1, we state under the heading “Population” whether the patients included in the primary study were already diagnosed by liver biopsy or were only clinically suspected to have NAFLD before inclusion in the primary study. This provides an idea on the severity of disease in the studied population, since patients generally undergo liver biopsy when disease is already more advanced. We checked back in the primary studies that included patients with suspected NAFLD to find out what this clinical suspicion was based on. Boyle 2019 and Luo 2018 do not specify on this point; in Bril 2019 all patients underwent proton MRS before the liver biopsy was taken. Because this information is not clear from all primary studies, we have decided not to make changes to Table 1.

Point 2: Lines 136-142 are unclear. What data used was taken only from published data, and which studies required additional data that was obtained by contacting the authors? This could be added as a separate table in the supplementary data section.

Response 2: Thank you for this comment, we agree that it should be clear which data was taken from published reports and which was additionally obtained by contacting the authors. We have added Supplementary Table 2 to clarify this point (see below) and we refer to this table in lines 213-214.

Supplementary Table 2. Additional data received from authors of primary publications

Study ID

Additional data requested?

What extra data received

Daniels 2019 [28]

Yes

2x2 data and cut-off for Pro-C3 alone (not ADAPT panel)

Boyle 2019 [27]

Not necessary

-

Huber 2019 [42]

Yes

2x2 data at two suggested cut-offs (15.6 and 21.3 ng/mL)

Luo 2018 [41]

Not necessary

-

Nielsen, Leeming 2021 [29]

Not necessary

-

Bril 2019 [40]

Yes

2x2 data for patients with NAFLD only, excluding the “No NAFLD” group

Knöchel 2021 [44]

Yes

Confirmation of exclusion criteria and no overlapping patient groups with other included studies; 2x2 data at two suggested cut-offs (15.6 and 21.3 ng/mL)

Erhardtsen 2021 [43]

Not necessary

-

Point 3: Reference 44 is an abstract, and therefore, this is not peer-reviewed published information. The data could be completely removed from the meta-analysis, or if included, a statement could be added indicating that the removal of these data does not affect the results shown in the Figures as well as the conclusions of the meta-analysis.

Response 3: It is indeed a valid point that non peer-reviewed information may be less reliable. Therefore, we have conducted an additional sensitivity analysis by removing the data from this abstract from our meta-analysis as suggested.

For significant fibrosis, the AUC remained 0.81; sensitivity changed from 0.68 to 0.62 and specificity changed from 0.79 to 0.81.

For advanced fibrosis, the AUC remained 0.79; sensitivity changed from 0.72 to 0.70 and specificity changed from 0.73 to 0.76.

We conclude that removal of the data from the abstract does not majorly affect the results as shown in the figures or the conclusions of this meta-analysis.

We have adapted the manuscript in lines 188-189 and 290-293 to describe this additional sensitivity analysis.

Reviewer 2 Report

REVIEW

Systematic Review with Meta-analysis: Diagnostic accuracy of Pro-C3 for hepatic fibrosis in patients with Non-Alcoholic Fatty 3 Liver Disease

GENERAL OVERVIEW

Topic is hot as described in the Intro. Fibrosis is the No.1 driving force behind the prognosis of NAFLD patients. Currently, its non-invasive assessment for a real-life clinical practice is the focus of uppermost interest and as of yet a need not copmletely met.

Invited to the author list were world-renowned experts which is adding considerably to the quality and credibility of the paper.

Language / readability: excellent

INTRO

Authors provide brief, sound and up-to-the point introduction in order to understand their aims. Intro has touched upon all the important issues connected with non-invasive diagnosis of fibrosis except transien elastography (and related such as ARFI, SWE etc): although not being the same category, they belong to the broader group of noninvasive diagnostic modalities of fibrosis and I recommend adding a brief mention of their role and that they are not the whole answer either.

Authors mention the role of markers of fibrosis for a real-life clinical practice as well as for a scientific studies (Lines 54+).

METHODS

2.1. The search of the literature (described as newly developed) fulfills the criteria needed to encompass a time-space necessary for hitting the target (Aim).

2.2. Eligibility criteria of the studies were meticulous and set the proper scene for analysis.

2.3., 2.6. – No comment

RESULTS

Flowchart and final yield confirm suspected high quality of MA’s methodology.

Line 199: since it is not stated in the Table 1, please provide in the text exact percentage of Asian patients in the two mentioned studies

I have no comments as to the credibility of results.

DISCUSSION

Main results: I suggest continuing the sentence (Line 300) with a few words on the suggested position of the test in the existing toolbox: Do authors recommend Pro-C3 as an adjunct-, or an alternative to ELF and FIB-4 (akin to the paragraph 4.4.).

Test performance - fibrosis. Interesting / inspiring area touching upon a dynamic component of fibrosis. Line 320. I suggest adding here a thought (from the lierature) on relative performance of fibrosis by histology vs by Pro-C3 (or other noninvasives) against the prognosis: which one of them is associated with worse survival, or other hard endpoint (as alluded to on Lines 378+ in terms of future directions).

Availability (line 374+): Please add also a comment on the price – at least relative to other diagnostic methods.

Strengths, limitations, future directions: No comment

Reviewer 3 Report

Review report

Manuscript ID: biomedicines-1472745

Title: Systematic Review with Meta-analysis: Diagnostic accuracy of Pro-C3 for hepatic fibrosis in patients with Non-Alcoholic Fatty Liver Disease

The systematic review and meta-analysis by A. Linde Mak et al. addresses the performance of Pro-C3 in detecting of hepatic fibrosis in patients with Non-alcoholic Fatty Liver Disease.

The manuscript is very interesting and the results of presented systematic review and meta-analysis may be very useful for clinicians in their practice. I appreciate the authors' effort to research and preparation of the manuscript, but some issues should be improved before consideration the text in publication in “Biomedicines” journal.

My decision: minor revision

Detailed comments and suggestions:

  1. Abstract: line 33-34: the sentence (“Two researchers independently…”) may be removed without the lowering the value of the abstract;
  2. Abstract and Literature search: the time frame of records identified in database searching should be indicated;
  3. The title of Table 1: “Characteristics of included studies” fits better than “Study characteristics of included titles”
  4. SROC curves (Fig. 2 and Fig. 3): the captions under the chart X axis should be replaced by “false positive rate”;

Author Response

Thank you for your positive remarks about our manuscript. Below here, we respond to the mentioned suggestions point by point.

Point 1: Abstract: line 33-34: the sentence (“Two researchers independently…”) may be removed without the lowering the value of the abstract;

Response 1: Thank you for this suggestion to make the abstract brief and comprehensible. We have removed the suggested sentence from the abstract (line 33).

Point 2: Abstract and Literature search: the time frame of records identified in database searching should be indicated;

Response 2: Indeed, we have added a sentence to clarify that we have included all records to date that fulfilled the search criteria: “All available records that fulfilled the search criteria to that date were screened.” (line 121-122).

Point 3: The title of Table 1: “Characteristics of included studies” fits better than “Study characteristics of included titles”

Response 3: Thank you for this accurate observation, we agree with your suggestion and have changed the title of Table 1 (line 222).

Point 4: SROC curves (Fig. 2 and Fig. 3): the captions under the chart X axis should be replaced by “false positive rate”;

Response 4: We have replaced the X axis captions (“1 – Specificity”) by “False positive rate” in Figures 2 and 3 (lines 242 and 266).

Round 2

Reviewer 1 Report

All points addressed. No further comments.